# Impaired Gut–Systemic Signaling Drives Total Parenteral Nutrition-Associated Injury

**DOI:** 10.3390/nu12051493

**Published:** 2020-05-20

**Authors:** Miguel Guzman, Chandrashekhara Manithody, Joseph Krebs, Christine Denton, Sherri Besmer, Pranjali Rajalakshmi, Sonali Jain, Gustavo Adolfo Villalona, Ajay Kumar Jain

**Affiliations:** 1Department of Pathology at Saint Louis University School of Medicine, SSM Cardinal Glennon Hospital, 1465 South Grand Blvd., St. Louis, MO 63104, USA; miguel.guzman@health.slu.edu (M.G.); sherri.besmer@health.slu.edu (S.B.); 2Department of Pediatrics at Saint Louis University School of Medicine, SSM Cardinal Glennon Hospital, 1465 South Grand Blvd., St. Louis, MO 63104, USA; chandrashekhara.manithody@health.slu.edu (C.M.); joseph.krebs@health.slu.edu (J.K.); christine.denton@health.slu.edu (C.D.); pranjali.rajalakshmi@slu.edu (P.R.); sonalij23@gmail.com (S.J.); 3Department of Surgery, Saint Louis University School of Medicine, 1402 South Grand Blvd. St. Louis, MO 63104, USA; villalog@icloud.com

**Keywords:** parenteral nutrition, liver disease, gut injury, gut–systemic crosstalk, hepato-biliary receptors and transporters

## Abstract

Background: Total parenteral nutrition (TPN) provides all nutritional needs intravenously. Although lifesaving, enthusiasm is significantly tempered due to side effects of liver and gut injury, as well as lack of mechanistic understanding into drivers of TPN injury. We hypothesized that the state of luminal nutritional deprivation with TPN drives alterations in gut–systemic signaling, contributing to injury, and tested this hypothesis using our ambulatory TPN model. Methods: A total of 16 one-week-old piglets were allocated randomly to TPN (*n* = 8) or enteral nutrition (EN, *n* = 8) for 3 weeks. Liver, gut, and serum were analyzed. All tests were two-sided, with a significance level of 0.05. Results: TPN resulted in significant hyperbilirubinemia and cholestatic liver injury, *p* = 0.034. Hepatic inflammation (cluster of differentiation 3 (CD3) immunohistochemistry) was higher with TPN (*p* = 0.021). No significant differences in alanine aminotransferase (ALT) or bile ductular proliferation were noted. TPN resulted in reduction of muscularis mucosa thickness and marked gut atrophy. Median and interquartile range for gut mass was 0.46 (0.30–0.58) g/cm in EN, and 0.19 (0.11–0.29) g/cm in TPN (*p* = 0.024). Key gut–systemic signaling regulators, liver farnesoid X receptor (FXR; *p* = 0.021), liver constitutive androstane receptor (CAR; *p* = 0.014), gut FXR (*p* = 0.028), G-coupled bile acid receptor (TGR5) (*p* = 0.003), epidermal growth factor (EGF; *p* = 0.016), organic anion transporter (OAT; *p* = 0.028), Mitogen-activated protein kinases-1 (MAPK1) (*p* = 0.037), and sodium uptake transporter sodium glucose-linked transporter (SGLT-1; *p* = 0.010) were significantly downregulated in TPN animals, whereas liver cholesterol 7 alpha-hydroxylase (CyP7A1) was substantially higher with TPN (*p* = 0.011). Conclusion: We report significant alterations in key hepatobiliary receptors driving gut–systemic signaling in a TPN piglet model. This presents a major advancement to our understanding of TPN-associated injury and suggests opportunities for strategic targeting of the gut–systemic axis, specifically, FXR, TGR5, and EGF in developing ameliorative strategies.

## 1. Introduction

Parenteral Nutrition (PN) is the process of intravenously administering nutrition to patients who are unable to meet nutritional requirements via enteral feeding. The key components of food—protein, carbohydrate, and fats—are provided intravenously, along with essential vitamins, minerals and micronutrients [1,2]. When nutritional needs are entirely met via such therapy, the process is called total parenteral nutrition (TPN). TPN, remains a critical modality of nutrition delivery throughout the world for neonatal, pediatric, as well as adult patients. Its use has grown exponentially over the past few decades [3,4,5].

Despite the lifesaving benefits of TPN, its use is associated with significant liver injury [6,7]. Several recent studies have also noted significant gut mucosal atrophy in association with TPN [8,9,10]. Currently, there are no proven therapeutics to treat or prevent TPN-associated liver and gut injury, with multi-visceral organ transplant the only viable option for patients with advanced disease [5,11].

Emerging data from several studies evaluating liver diseases such as nonalcoholic steatohepatitis have highlighted the importance of the gut in modulating liver injury [12,13,14,15]. Gut signaling is known to be regulated by luminal content [14,16]. Interestingly, in clinical settings, TPN injury is minimal if some enteral nutrition is provided, confirming the important role of the gut in mediating TPN-associated liver injury. The term intestinal failure-associated liver disease (IFALD) is increasingly being used, replacing the acronym parenteral nutrition-associated liver injury (PNALD) to better characterize TPN-induced hepatic injury [17,18]. However, the exact role of the gut in modulating TPN injury remains unclear.

Because TPN results in a state of significant nutrient deprivation for the gut mucosa, we hypothesized that it is this state of luminal content deprivation during TPN therapy that induces alterations in gut-derived signaling, contributing to liver and gut injury.

We have developed a novel ambulatory TPN piglet model recapitulating human TPN delivery [1]. Using this model system, our aims for this study were to obtain crucial insights into mechanistic pathways driving the altered gut–systemic signaling during TPN, which could ultimately create the foundation for preventative strategies, therapeutic drug development, and an understanding of the intricate pathways altered with TPN. Our key surrogates for TPN-associated injury were changes in serum bilirubin level and gut atrophy, as modulated by gut–systemic signaling.

## 2. Materials and Methods

### 2.1. Animal Procurement, Surgery, and Nutrition Delivery

Saint Louis University (SLU) is a U.S. Department of Agriculture-registered research facility. This study was conducted upon approval by the University Institutional Animal Care and Use Committee (SLU no. 2357, U.S. Department of Agriculture registration number 43-R-011). It was performed in accordance with the *Guide for the Care and Use of Laboratory Animals*, as well as being compliant with the Animal Research Reporting of In Vivo Experiments (ARRIVE) Guidelines for Reporting Animal Research.

Sixteen one-week-old, farm raised, term neonatal pigs (piglets) were utilized for this study. Animals were obtained from a university-approved class A vendor. They were identified by ear tags. In accordance with university policies and guidelines, piglets were acclimatized during the first 72 h, being fed ad libitum, with close monitoring of their daily intake. All animals were kept in a thermally controlled environment for the entire study. 

Post acclimatization, the animals were taken to the operating room (OR) and, as previously described, jugular and duodenal catheters were placed The animals were then randomly allocated to receive enteral nutrition (EN) or TPN, as previously reported [1,19]. 

### 2.2. Nutrition

For the EN group, a swine replacement formula was provided enterally, as previously described. For the TPN group, a commercially available parenteral nutrition preparation and lipid was delivered via the jugular catheter, as previously described [1,20,21]. The nutritional constituents are included in Table 1. These formulations, as we have published before, conform to applicable doses in humans [19,20,22].

### 2.3. Animal Monitoring

The animals were monitored continuously by scheduled visits by the research staff, in accordance with the *Guide for the Care and Use of Laboratory Animals* as well as the Institutional Animal Care and Use Committee [23].

### 2.4. Animal Euthanasia and Tissue Collection

As described previously [1,8], the animals were euthanized after 3 weeks into the study. A blood sample was obtained just prior to euthanasia. A liver panel and lipid profile were analyzed by an automated analyzer at SLU core facility. Post euthanasia, the abdomen was surgically opened, and the liver removed in its entirety. The entire small intestine was removed and flushed with saline. The intestinal contents were extruded and weighed. Small segments of the liver, and distal as well as proximal small bowel were then sliced, measured, and weighed.

### 2.5. Histology

Segments of small intestine and fragments of liver, each measuring approximately 2–3 cm in greatest dimension, were fixed in buffered formalin (4%) for 24 h and then were stored in 70% ethanol at room temperature for 24 h. The tissue was subsequently processed, embedded in paraffin, and stained with hematoxylin and eosin (H&E). Liver tissue was additionally stained with Sirius red for evaluating fibrosis. Liver tissue also underwent staining for cytokeratin-7 and cluster of differentiation 3 (CD3). An automated upright microscopy system with light-emitting diode (LED) illumination for life sciences, Leica DM4000 B LED, was utilized, along with the Q-capture pro digital imaging software. 

#### 2.5.1. Histology Scoring

All slides were reviewed by light microscopy by the study pathologist who was blinded to the study arms. The following describes the specific stains as well as the methods utilized for quantification.

#### 2.5.2. Cytokeratin 7 (CK-7)

CK-7 is a low molecular weight cytokeratin that is expressed in the epithelial cells in the liver that have a bile duct phenotype. Therefore, CK-7 serves as a marker for ductular proliferation [24,25]. We performed CK-7 staining on the liver tissue of both enteral nutrition (EN) as well as TPN animals. All CK-7 immune-reactive cells (membranous and cytoplasmic pattern) were counted and reported in five non-overlapping high-power fields, which were then divided by the total number of cells in order to compute a final CK-7 score.

#### 2.5.3. Cluster of Differentiation 3 (CD-3)

CD-3 stains T cells, which are the predominant inflammatory cells seen in hepatic inflammation. Therefore, CD-3 was performed on the liver tissue as a marker of hepatic inflammation [26]. The number of positively stained cells was counted and divided by total number of cells in five non-overlapping high-power fields in order to compute a final CD3 score.

#### 2.5.4. Sirius Red Staining

Collagen provides structure as well as support to maintain healthy liver function; however, an increase in collagen deposition is a feature of hepatic injury and fibrosis [27], and is detected with Sirius red staining [28]. The amount of collagen in each slide was determined by calculating the ratio of collagen stained by Sirius red as a percentage of the entire sample.

#### 2.5.5. Liver Cholestasis Scoring

All liver H&E slides were reviewed by light microscopy for this analysis. Bile deposits indicative of cholestasis [29] were identified, and the cholestatic foci were then counted in three non-overlapping fields in order to create a final liver cholestasis score.

#### 2.5.6. Morphometric Analysis of Small Bowel Epithelium

The villous height as well as crypt depth were measured in five vertically well oriented villous–crypt columns in small bowel H&E slides in order to compute the villous-to-crypt ratio for animals in each group.

#### 2.5.7. Muscularis Mucosa Assessment

The distal gut slides were reviewed using a light microscope at 400× magnification in five non-overlapping fields to measure the thickness of the muscularis mucosa.

#### 2.5.8. Electron Microscopy

The liver samples from the EN and TPN animals were collected and fixed in 2.5% glutaraldehyde. The specimens were further processed into thin slices and viewed by transmission electron microscopy to study the ultrastructural changes. 

### 2.6. Tissue: RNA Extraction and Real Time PCR Analysis

RNA was extracted from the liver as well as gut tissue using a Sigma-Aldrich GenElute Mammalian Total RNA Miniprep Kit (RTN70-1KT) and Trizol (ThermoFisher, 15596018), respectively, as per the protocol. Isolated RNA (1ng) was then reverse-transcribed into complementary DNA utilizing a Verso complementary DNA (cDNA) Synthesis Kit (ThermoFisher, AB1453B). Primers for each transcript were designed using Integrated DNA Technologies (IDT) and validated (Table 2). Real-time quantitative polymerase chain reaction (RT-qPCR) was then performed in triplicate, utilizing the Bio-Rad CFX Connect Real-Time System. Relative mRNA was then calculated using the comparative threshold cycle method with beta-actin as an internal control.

### 2.7. Statistical Analysis

Graph Pad Prism version 7.03 software was used for statistical analysis. Descriptive statistics for outcomes were calculated as median and interquartile range (IQR). Pairwise Mann–Whitney *U* tests were performed for assessing serological markers, histology reads, and relative mRNA expression of the genes. All statistical tests were two-sided and used a significance level of 0.05.

### 2.8. Sample Size

At a power of 80% and significance level of 0.05, a sample size of eight piglets in each group allowed for an effect of 1.5 for an independent samples *t*-test, 1.3 for a Mann–Whitney *U* test, and proportions of 0.7 vs. 0.05 for a chi-squared Fisher’s exact test. Serum bilirubin and changes in villous to crypt (V/C) ratio, which are hallmarks for TPN-associated liver and gut injury, were used as surrogates for initial sample size calculation. In consultation with our university statistician, for our calculations, we used historical data of 5-10-fold changes in serum bilirubin [1,18,30,31], as well as a 2–3 fold reduction in the V/C ratio [1,22,29] with TPN to model the sample size calculation. From our previous work [1,7,8,20,21,32], we predicted that the actual effects across variables would be larger, thus reasonably obtainable with eight animals per group.

## 3. Results

### 3.1. Baseline Assessment and Weight Changes

A total of 16 animals were included in the study. At baseline, both the groups were comparable for age as well as weight. There were no statistical differences in daily weight gain in either the control or the TPN group over 3 weeks of treatment (*p* = 0.798). The median and IQR for daily weight gain for the control animals was 95.36 g (90.67–104.4), and for the TPN animals was 93.93 g (90.81–100.9).

### 3.2. Serum Bilirubin

A key marker for cholestatic liver injury with TPN therapy is an elevation in the serum bilirubin level. We report a significant elevation in serum bilirubin (Figure 1A) in animals on TPN (*p* = 0.034). The median and IQR for serum bilirubin for the control EN group was 0.15 (0.10–0.21) mg/dL and for TPN was 1.58 (0.17–5.45) mg/dL. We also measured total serum bile acids. Serum bile acids (Figure 1B), were significantly elevated in the TPN group (*p* = 0.029). The median and IQR for serum bile acids for the enteral control group was 6.20 (2.32–13) mg/dL and for TPN was 36.20 (8.28–62.6) mg/dL.

### 3.3. Bile Deposits and Hepatic Serology

Histologically, higher hepatic intra-parenchymal bile deposition was noted in animals on TPN (Figure 2A,B). For each animal, a hepatic cholestasis score was calculated on the basis of these cholestatic deposits. In line with the reported serum bilirubin results, hepatic cholestasis was significantly higher in the TPN animals, with the median and IQR being 10.56 (3.11–13.19) for the control group (Figure 2C) and 14.06 (12.89–20.61) for animals on TPN (*p* = 0.028). The cholestatic deposits were localized to the portal areas and were visualized both on H&E histology and confirmed with electron microscopy (EM). We did not note hepatocytic necrosis or acidophilic bodies with TPN either on H&E histology or EM (Figure 2D,E). The degree of steatosis was also not different between the groups. We also evaluated the serological markers for liver injury as well as the lipid profile. Paralleling a lack of hepatocytic injury with TPN, as noted on histology, no statistical differences in serum alanine aminotransferase (ALT) level (*p* = 0.083) or serum cholesterol (*p* = 0.065) were noted between the groups.

### 3.4. Hepatic Immunohistochemistry

Downstream markers of liver injury were examined by quantification of fibrosis and hepatocellular turnover. Although collagen deposition resulting in fibrosis has been noted to occur with TPN, objective quantification with Sirius red staining did not reveal statistically significant hepatic fibrosis (*p* = 0.476) in comparison to the controls in our study. In order to characterize differences in the inflammatory infiltrate between the two groups, we performed CD3 immunohistochemistry (Figure 3A–C). We noted a significantly greater inflammatory infiltrate, as demonstrated by the number of CD3-positive cells in animals on TPN (*p* = 0.021). The median and IQR for CD3 for control was 9.83 (7.33–14.42) and for TPN was 22.67 (11.0–27.67).

We also quantified bile ducts per portal track using CK-7 immune staining. Stains for CK-7 (Figure 3D–F) did not show any statistical difference in number of bile ducts at the portal triads (*p* = 0.672) among the groups.

### 3.5. Gut Morphology and Histology

Although the small bowel in animals on TPN was thin and friable on gross inspection, there was preservation of gut morphology in animals receiving enteral nutrition.

### 3.6. Villous/Crypt Ratio

We report significant villous atrophy in animals on TPN compared to those receiving enteral nutrition (Figure 4A–C). The median and IQR for the V/C for control was 2.71 (2.18–3.1) and for TPN was 1.92 (1.66–2.51), *p* = 0.028.

Bowel weight: After animal sacrifice, upon identification of the ileo-cecal valve, a 20 cm segment of the bowel was removed proximal to the valve and weighed in each animal for consistency across study arms. In order to compare bowel growth among the two groups, we then calculated the weight per centimeter of the small bowel and determined the “gut mass”. Supporting the changes noted grossly, a significant reduction in gut mass was noted in the TPN group compared to the enteral nutrition animals. The median and IQR for gut mass and for control EN group was 0.46 (0.30–0.58) g/cm compared to TPN, which was 0.19 (0.11–0.29) g/cm, *p* = 0.024 (Figure 4D). In fact, we noted that the gut mass reduction was accompanied by a significantly decreased thickness of the muscularis mucosa for the animals receiving TPN. The median and IQR for muscularis mucosa thickness for EN group was 28.72 (26.27–32.88) microns compared to TPN, which was 23.41 (20.93–25.48) microns, *p* = 0.010 (Figure 4E–G).

### 3.7. Gut-Systemic Signaling—Key Gut and Hepatobiliary Receptors and Transporters

Gut farnesoid X receptor (FXR)–cholesterol 7 alpha-hydroxylase (CyP7A1): In clinical settings, TPN-associated injury is minimized if some enteral nutrition is also provided [33,34]. In fact, studies in both cell culture as well as animal models show that activation of the recently characterized, gut nuclear receptor farnesoid X receptor (FXR) in intestinal epithelial cells [35,36,37] regulates hepatic bile acid synthesis via the rate-limiting step cholesterol 7 alpha-hydroxylase (CyP7A1) [38]. We postulated that TPN-associated liver injury is due to inadequate gut FXR activation due to a lack of luminal nutrients, as occurs during TPN therapy. We thus measured mRNA expression of multiple key gut and hepatic receptors and transporters that are known to drive the gut–systemic signaling.

We noted significantly reduced expression of the gut FXR mRNA expression in animals on TPN (Figure 5A) compared to those on enteral nutrition (*p* = 0.028). FXR suppresses CyP7A1 expression, and we found an almost fivefold increase in expression of hepatic CyP7A1 in TPN animals (Figure 5D) in the setting of low FXR expression. It has been noted that gut FXR activation stimulates production of the metabolic growth factor peptide fibroblast growth factor-19 (FGF19) [39,40]. However, although FGF19 expression was numerically lower in TPN animals, this difference did not reach statistical significance. The median and IQR relative FGF19 for EN group was 0.11 (0.007–2.55) compared to TPN 0.53 (0.16–3.68), *p* = 0.282. In line with these results, we noted a significant reduction in hepatic FXR (*p* = 0.021), which is known to regulate both intrahepatic bile acid metabolism and hepatic inflammation [41,42], as well as the key hepatic endobiotic detoxifier, the constitutive androstane receptor (CAR) [43,44], a known member of the nuclear receptor superfamily (*p* = 0.01) (Figure 5E,F).

Gut G-coupled bile acid receptor (TGR5) and epidermal growth factor (EGF): It has been postulated that intestinal growth is regulated by the enteral bile acid-regulated luminal G protein-coupled receptor TGR5 [45,46,47], which is known to be highly localized to crypts. Gut-derived signaling secondary to luminal epidermal growth factor (EGF) regulation also controls gut mucosal and villous proliferation [48,49]. We noted significant reduction in expression of both TGR5 (*p* = 0.003) as well as EGF (*p* = 0.016) mRNA in TPN animals (Figure 5B,C). If fact, there was an almost three- and five-fold reduction in TGR5 and EGF expression, respectively, with TPN.

We next evaluated the expression of the key gut growth peptides in gut lysate, which are known to be regulated via TGR5 and EGF. As a surrogate, we evaluated the precursor to glucagon-like peptide [19,50,51]. In parallel to the decreased gut EGF and TGR5 expression, likely driven by a lack of luminal activation, we noted its significant reduction in animals on TPN (*p* = 0.031).

We proposed that TPN-associated injury is a consequence of impaired luminal signaling. We noted gut atrophy in TPN animals. In addition, we also noted that in animals receiving TPN, there was a significant reduction in the expression of organic anion transporter (OAT), which is a key luminal bile acid transporter (*p* = 0.028). Gut expression of mitogen-activated protein (MAP) kinase, which represents a key signal transduction pathway, was significantly reduced (almost twofold) with TPN (*p* = 0.037). TPN also resulted in a marked, almost eightfold reduction (*p* = 0.010), in the expression of the major luminal sodium-glucose linked transporter (SGLT-1) in comparison to control EN animals (Figure 5G–I).

## 4. Discussion

Parenteral nutrition is the accepted method of providing all nutritional needs intravenously. When nutritional needs are entirely met via such intravenous therapy, the process is referred to as TPN. Such therapy remains an essential modality of nutrition delivery throughout the world for neonatal, pediatric, and adult patients [30]. Its use has grown exponentially over the past few decades. However, TPN, despite being a lifesaving therapy, is marred by major complications of liver injury and significant gut atrophy [10,32,52]. Although injury mitigation by infection control [53], lipid minimization, avoidance of predominant soy based lipids [54,55], cycling of TPN [56], and enteral nutrition have been proposed [57], the etiology for the detrimental effects of TPN remain largely unknown.

Although several mechanisms have been postulated, this study explores whether or not there is an impairment of gut-derived signals, due to a lack of luminal nutrients during TPN, which drive TPN-associated injury. This idea is also supported by clinical practice, where TPN related injury is known to be mitigated if some enteral nutrition can be provided [33,58]. We thus hypothesized that alteration of gut–systemic signaling may play a critical role in TPN injury.

In this study, we used a novel ambulatory piglet TPN model. We utilized this model to study human physiology because of the extensive homology of the liver and gastrointestinal tract of the pig [30,59] to humans in both form and function [59,60,61,62]. This model is therefore, highly relevant to human pediatric and adult TPN injury [1,8,63].

Utilizing this system, we described a comprehensive histo-pathological assessment of gut and liver pathology. Lack of enteral feeding in our animals on TPN disrupted the normal luminal signaling. Our study highlighted the concept that TPN injury might not only be secondary to a toxic insult due to the components of TPN, but rather likely a result of the state of prolonged lack of luminal signaling as associated with TPN therapy.

In clinical settings, IFALD is primarily diagnosed by elevated serum bilirubin and aminotransaminases. Its pathology is characterized by cholestasis, with reports suggesting a variable degree of fibrosis, steatosis, bile ductular changes, as well as hepatic inflammation. Usually these changes have been reported to occur after several weeks of TPN therapy. Although some of this variability may be related to the duration of TPN use, as well as differing lipid emulsions [64,65,66], our study presents the idea that the initial responses to TPN might be driven by an alteration of gut-derived signaling with subsequent bilirubin, bile acid, and other toxic metabolite accumulation, which then evolves into hepatocellular injury.

Thus, although there was cholestatic liver disease, by both histology and serum bilirubin elevation, we did not note ALT elevations, bile ductular proliferation, or histologic hepatocellular injury in our animals during our 3 week study. We did, however, note significant alterations in key hepatobiliary receptors and transporters as part of the gut–liver crosstalk, as well as key components of bile acid signaling pathway and metabolism that drives liver injury. Prior studies have suggested that gut FXR activation in the intestinal epithelial cells [35,36] can regulate hepatic bile acid synthesis via modulation of CyP7A1 (cholesterol 7 alpha-hydroxylase, the rate limiting step of bile acid synthesis) [38]. We noted significant elevation of CyP7A1 and bile acids in animals on TPN, indicating excessive bile acid production untampered by a lack of gut signaling, which normally occurs in enteral animals. Other noted alterations in key hepatobiliary receptors and transporters in animals on TPN influence bile acid transport and excretion. This results in bile acid and possibly other toxic metabolite accumulation over time, further exacerbating injury.

We thus predict that longer durations of TPN would result in classical hepatocellular injury, as well as ALT elevations characteristic of clinical TPN associated injury, and this will be the focus of further studies.

It is important to note that although, in clinical settings, ursodeoxycholic acid (UDCA) has been used in patients on TPN with liver disease with inconsistent results [6,67,68], UDCA has minimal activity for FXR [35,69], and thus cannot activate the gut–systemic signaling pathway, which possibly explains this lack of response.

While mucosal atrophy and a reduction in the villous to crypt ratio has been previously reported [70,71], our other novel finding is the noted significant reduction in thickness of muscularis mucosa as part of the gut mass differences with TPN. Although not the focus of this study, it is possible that gut motility differences noted with TPN are a surrogate of such changes in the muscle layers. Additionally, the observation of a reduction in gut mRNA expression with TPN for several receptors and transporters could be a derivative of changes in gut structure, which may require additional studies.

In summary, this study shows that the likely initial event with TPN is an alteration of the gut–systemic signaling, which we postulate results in bile acid and other toxin accumulation and subsequent hepatocellular dysfunction as TPN is continued for longer duration. This, we believe, presents a major shift in our understanding of TPN associated injury. 

Studies using tissue specific knock-out models and receptor agonistic and antagonist trials could help further clarify mechanistic signaling pathways. Additionally, further studies targeting luminal receptor activation that replicate signaling during regular enteral nutrition could greatly help in the development of diagnostic and therapeutic strategies to mitigate against TPN-associated complications.

## Figures and Tables

**Figure 1 nutrients-12-01493-f001:**
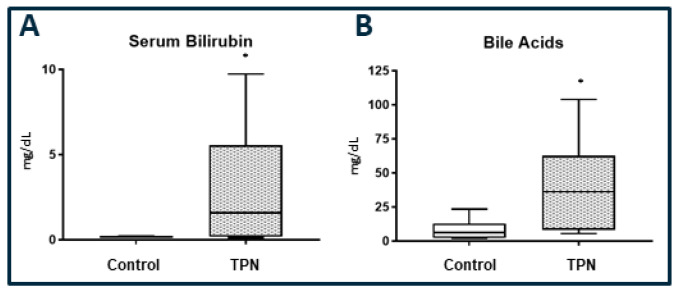
(**A**): Serum bilirubin. (B) Total serum bile acids. Significantly high serum bilirubin and bile acids levels were found with total parenteral nutrition (TPN) compared to enteral nutrition (control). The figure shows box and whisker plots in which boxes represent the 25–75th percentile, and the center lines represent median values. A pairwise Mann–Whitney *U* test was conducted to determine the *p*-value. All tests were two-sided, using a significance level of 0.050.

**Figure 2 nutrients-12-01493-f002:**
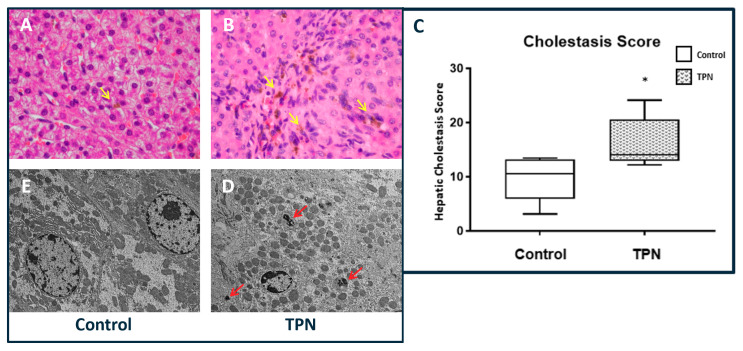
Hepatic cholestasis. (**A**,**B**) Hematoxylin and eosin (H&E) stain, 40× magnification showing increased cholestasis in TPN animals (yellow arrows). (**C**) Cholestasis score was significantly higher with TPN. The figure shows box and whisker plots in which boxes represent the 25–75th percentile, and center lines represent median values. A pairwise Mann–Whitney *U* test was conducted to determine *p*-value. All tests were two-sided, using a significance level of 0.05. (**D**,**E**) Electron microscopy at 5000× magnification showing bile deposits (red arrows) in TPN-treated animals but not in controls.

**Figure 3 nutrients-12-01493-f003:**
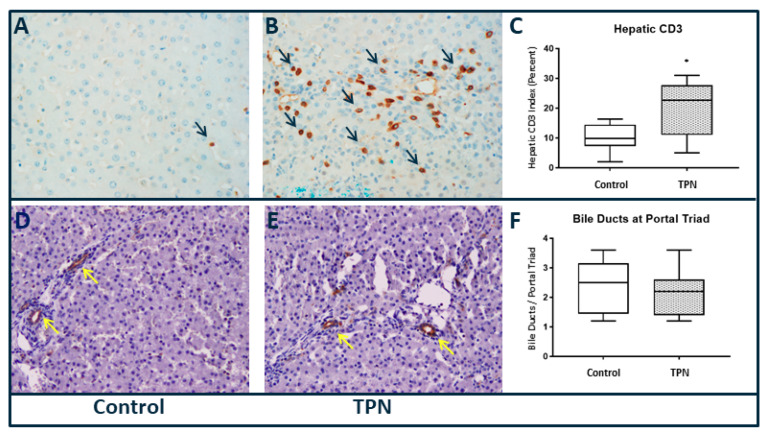
Liver immunohistochemistry, 40× magnification (**A**–**C**) cluster of differentiation 3 (CD3) and (**D**–**F**) cytokeratin 7 (CK-7). Significantly higher CD3 staining was found with TPN. No differences in CK-7 staining of bile ducts were found. Histology scoring was performed, and a pairwise Mann–Whitney *U* test was conducted to determine *p*-value. All tests were two-sided, using a significance level of 0.05.

**Figure 4 nutrients-12-01493-f004:**
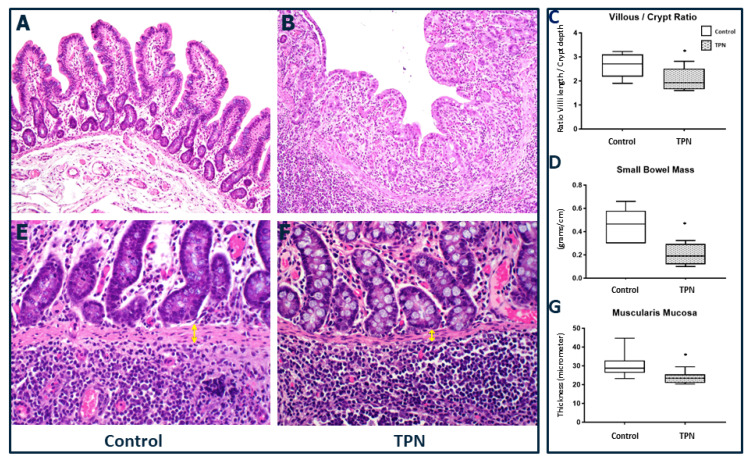
Small bowel histology. (**A**–**C**) Villous/crypt ratio (20× magnification). Significant blunting of villi was found with TPN. (**D**) Gut mass of small bowel was lower with TPN. (**E**–**G**) Muscularis mucosa thickness (40× magnification) was decreased in animals treated with TPN (yellow arrows). The figure shows box and whisker plots in which boxes represent the 25–75th percentile and center lines represent median values. A pairwise Mann–Whitney *U* test was conducted to determine the *p*-value. All tests were two-sided, using a significance level of 0.05.

**Figure 5 nutrients-12-01493-f005:**
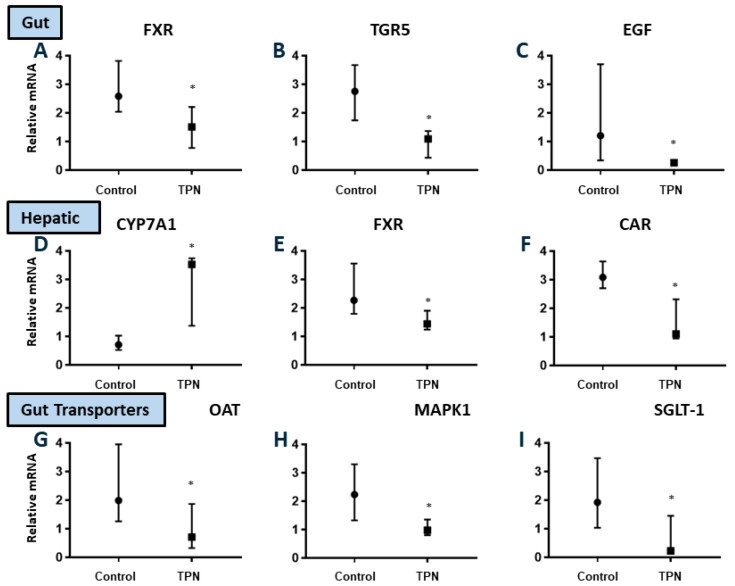
Relative mRNA expression for key bile acid-regulated hepatobiliary receptors and transporters in the gut and liver (**A**–**I**). Significant downregulation of gut FXR, TGR5, and EGF, as well as hepatic FXR and CAR in TPN animals. Upregulation of hepatic CyP7A1 was found with TPN. There was significant downregulation of gut OAT, Mitogen-activated protein kinases-1 (MAPK1), and SGLT-1 with TPN. The figures show median and interquartile values. Filled circles represent EN and filled boxes represent TPN. A pairwise Mann–Whitney *U* test was conducted to determine the *p*-value. All tests were two-sided, using a significance level of 0.05.

**Table 1 nutrients-12-01493-t001:** Nutritional constituents.

Total Parenteral Nutrition	Enteral Nutrition
Ingredients: Leucine, Isoleucine, Valine, Lysine, Phenylalanine, Histidine, Threonine, Methionine, Tryptophan, Alanine, Arginine, Glycine, Proline, Serine, Tyrosine, Sodium, Potassium, Magnesium, Calcium, Acetate, Chloride, Phosphate, Dextrose, and Intralipid (Ingredients: Soybean Oil, Egg Yolk Phospholipids, Glycerin and Water).	Ingredients: Dried Whey Protein Concentrate, Animal Plasma, Animal and Vegetable Fat preserved with beta hydroxy acid (BHA), Dried Lactose, Lecithin, Dicalcium Phosphate, Magnesium Sulfate, Manganese Sulfate, Ferrous Sulfate, Zinc Sulfate, Cobalt Sulfate, Copper Sulfate, Calcium Iodate, Sodium Selenite, Vitamin A Acetate, d-Activated Animal Sterol, Vitamin E Supplement, Menadione Dimethylpyrimidinol Bisulfite, Choline Chloride, Riboflavin Supplement, Calcium Pantothenate, Niacin Supplement, Vitamin B12 Supplement, Biotin, Ascorbic Acid, Yucca Schidigera Extract, and Natural and Artificial Flavors.

**Table 2 nutrients-12-01493-t002:** Primer sequences.

Primer		Sequence
Fibroblast growth factor 19: FGF19	Forward	ACACCATCTGCCCGTCTCT
	Reverse	CCCCTGCCTTTGTACAGC
Farnesoid X receptor (FXR)	Forward	ACATTCCTCATTCTGGGGCTTT
	Reverse	TTTCGGGGTCTTACTCCTTACA
Cholesterol 7 alpha-hydroxylase (CyP7A1)	Forward	AGGGTGACGCCTTGAATTT
	Reverse	GGGTCTCAGGACAAGTTGGA
Constitutive androstane receptor (CAR)	Forward	CCGCCATATGGGCACTATGT
	Reverse	GCGAAATGCATGAGCAGAGA
G protein-coupled receptor TGR5	Forward	CCATGCACCCCTGTTGCT
	Reverse	GGTGCTGTTGGGTGTCATCTT
Epidermal growth factor (EGF)	Forward	ACTACTACAGGACTCAGAAG
	Reverse	CCTGATACCACTCACATCTC
Organic anion transporter (OAT)	Forward	GAAAATGCCGAGAAGATGG
	Reverse	CAAGCGTCGTAATCTTTGG
Mitogen-activated protein (MAP) kinase	Forward	CTACACCAACCTCTCCTAC
	Reverse	GTAGGTCTGATGCTCAAATG
Sodium uptake transporter sodium glucose-linked transporter (SGLT-1)	Forward	GGCTGGACGAAGTATGGTGT
	Reverse	ACAACCACCCAAATCAGAGC
Beta actin	Forward	GGACCTGACCGACTACCTCA
	Reverse	GCGACGTAGCAG AGCTTCTC

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
