# Peer review of "Impaired Gut–Systemic Signaling Drives Total Parenteral Nutrition-Associated Injury"

_nutrients, 2020, doi:10.3390/nu12051493_

Round 1
Reviewer 1 Report
Thanks for the opportunity to review the work titled "Impaired gut-systemic signalling drives Total 2 Parenteral Nutrition associated injury"This work has merit and has nicely demonstrated the translational mechanism of gut-signalling in TPN associated gut and liver injury. There are some grammatical errors to be corrected.
I have a few comments for the authors to consider: Major comments:
1. Can the author clarify the sample size calculation - which outcome and effect were used for sample size calculation?
2. How did the author select the portion of the gut for gut-mass evaluation to avoid sampling bias?
3. The onset of gut atrophy in piglet started from day 3 (Conour 2002). Can the author explain what is the rationale of sampling only after 3 weeks of TPN? Minor comments:
1 . ESPEN guideline provides recommendations on IFALD which include infection control, enteral intake, cycled PN, avoid overfeeding and limit soybean-oil based lipid to < 1g/kg/d, and should be included in introduction and discussion
2. The author mentioned there is no significant difference in the value of ALT. This should be clearly reported under the result section
3. Composition and content of EN and PN should be better described
4. "median and IQR for the V/C for control was 2.71 (2.18-3.1) and for TPN was 1.92 (1.66-2.51)" - is this statistically significant?
5. "significant reduction in the gut mass was noted in the TPN 245 group compared to the enteral nutrition animals (p=0.024)" - please specify the gut mass between the 2 groups?
6. "..FGF19 expression was numerically lower in TPN animals, the difference did not reach statistical significance (p=0.2)." - again, please report the value in both groups for comparison
7. The author mentioned collagen deposition was performed using Sirius red staining. The result was not described under the result section.
Author Response
Apr 30, 2020
Editorial Office, Nutrients,
Dear Reviewers,
Thank you once again for considering our manuscript entitled, ‘Impaired gut-systemic signaling drives Total Parenteral Nutrition associated injury’.
Thank you very much for thoroughly critiquing our manuscript. Your recommendations have been extremely helpful. We humbly believe that we have addressed all the critique from reviewers.
Please find below a point by point response to the reviewer comments. Each critique is followed by ‘Response’ in red indicating our action.
We have updated the text in the manuscript per reviewer critique. The entire manuscript has been checked for syntax and grammatical errors. Several new references have been added. Figure legends have been updated. A new table has been added. Additionally, all figures and tables have been revised per journal formatting.
If any section or subsection needs further revision, please let us know and we shall promptly comply.
Sincerely,
Ajay Kumar Jain, MD
******** REVIEWER 1 ********
Thanks for the opportunity to review the work titled "Impaired gut-systemic signaling drives Total 2 Parenteral Nutrition associated injury"
Critique:
This work has merit and has nicely demonstrated the translational mechanism of gut-signaling in TPN associated gut and liver injury. There are some grammatical errors to be corrected.
Response: Thank you for the critique and recognizing our work. We have checked the entire manuscript for grammatical and syntax errors and the manuscript has undergone English editing.
Critique:
I have a few comments for the authors to consider: Major comments: Can the author clarify the sample size calculation - which outcome and effect were used for sample size calculation?
Response: Thank you for the critique. We have updated the section on sample size calculation and included the statement, ‘Serum bilirubin and changes in villous to crypt ratio, which are hallmarks for TPN associated liver and gut injury were used as surrogates for initial sample size calculation’. From our previous work we believe the actual effects across variables will be larger, thus reasonably obtainable with 8 animals per group.
Critique:
How did the author select the portion of the gut for gut-mass evaluation to avoid sampling bias?
Response: Thank you for this critique. We recognize that this was not clearly detailed and we have updated the manuscript as follows. ‘After animal sacrifice, upon identification of the ileo-cecal valve, a 20 cm segment of the bowel was removed proximal to the valve and weighed in each animal for consistency across study arms. In order to compare bowel growth among the groups we then calculated the weight per centimeter of the small bowel as an estimate of the ‘gut mass’.
Critique:
The onset of gut atrophy in piglet started from day 3 (Conour 2002). Can the author explain what is the rationale of sampling only after 3 weeks of TPN?
Response: Thank you for this critique. As the review has noted, prior publications have reported that gut atrophy can occur as early as 3 days upon initiation of TPN, however liver injury may not appear in this short period of time, specifically significant bilirubin elevation, which is a hallmark of TPN liver injury.
In clinical practice TPN associated liver and gut injury generally occurs upon several weeks of TPN use. While tethered animal models for longer duration TPN delivery have been used, they result in animals stress from tethering and result in alteration of hepato-biliary receptors. This manuscript is thus unique as it characterizes TPN injury in an ambulatory (non-tethered) animal model which truly can discern and accurately ascertain the impact of TPN on liver and gut injury. We limit to the 3 weeks durations as that is sufficient to get the spectrum of injury with TPN including changes in gene expression and downstream signaling as attested by our results and their relevance to gut-systemic signaling, which is the focus of this manuscript.
Critique: Minor comment:
ESPEN guideline provides recommendations on IFALD which include infection control, enteral intake, cycled PN, avoid overfeeding and limit soybean-oil based lipid to < 1g/kg/d, and should be included in introduction and discussion.
Response: Thank you for this critique. While the focus of this study is to explore the novel idea of alteration in gut-systemic signaling as a driver for TPN associated injury, the authors recognize the mitigation strategies reported by ESPEN as indicated in the minor comment by the reviewer and these have been included in the discussion. An additional 5 references have been added.
Critique: Minor comment:
The author mentioned there is no significant difference in the value of ALT. This should be clearly reported under the result section.
Response: Thank you for this critique. The ALT results have been reported in the discussion section, including a p value.
Critique: Minor comment:
Composition and content of EN and PN should be better described.
Response: Thank you for this critique. A new table has been added with the nutritional constituents of TPN and enteral nutrition. The method section has been updated to reflect this addition.
Critique: Minor comment:
"median and IQR for the V/C for control was 2.71 (2.18-3.1) and for TPN was 1.92 (1.66-2.51)" - is this statistically significant?
Response: Thank you for this critique. Yes, this was statistically significant with a p value of 0.028. This has been added to the results section.
Critique: Minor comment:
"significant reduction in the gut mass was noted in the TPN group compared to the enteral nutrition animals (p=0.024)" - please specify the gut mass between the 2 groups?
Response: Thank you for this critique. While this was included in the result section, it has been moved next result description and the p value added. We have thus stated that, ‘Supporting the changes noted grossly, a significant reduction in the gut mass was noted in the TPN group compared to the enteral nutrition animals. The median and IQR for gut mass and for control EN group was 0.46 (0.30-0.58) g/cm compared to TPN which was 0.19 (0.11-0.29) g/cm, p=0.024 (Figure 4D). In fact, we noted that the gut mass reduction was accompanied by a significantly decreased thickness of the muscularis mucosa for the animals receiving TPN. The median and IQR for muscularis mucosa thickness for EN group was 28.72 (26.27-32.88) microns compared to TPN which was 23.41 (20.93-25.48) microns, p=0.01 (Figure 4E-G).’
Critique: Minor comment:
"..FGF19 expression was numerically lower in TPN animals, the difference did not reach statistical significance (p=0.2)." - again, please report the value in both groups for comparison.
Response: Thank you for this critique. The median and interquartile range for each group has been added for FGF19 in the results section.
Critique: Minor comment:
The author mentioned collagen deposition was performed using Sirius red staining. The result was not described under the result section.
Response: Thank you for this critique. While we did report fibrosis in the results section (which occurs upon collagen deposition), the sentence has been re-written for greater clarity as, ‘on objective quantification with Sirius Red staining, hepatic fibrosis did not reach statistical significance (p=0.476) in comparison to controls in our study’

Reviewer 2 Report
The study investigates the underlying reasoning of side effects of total parenteral nutrition on the liver and gut by utilising pig models. The study design seems scientifically accurate and the manuscript was satisfactorily well written. Moreover, the authors report very interesting findings regarding the alterations in gut-derived signaling, which will potentially interest the readers.
I have no major concerns and recommend for acceptance after correcting the following minor issues:
- What are the sizes of the histology images? Scale bars would be helpful for a better understanding.
- Please use consistent decimals for the p-values reported (preferably 3 decimals for all)
- What do the asterisks in the boxplot figures mean? I suppose they represent significant differences between the groups. If so, what does the asterisk on the left (near y-axis) mean in Figure 2C?
- No need to explain details of box plots in every figure e.g., "the figure shows box and whisker plots in which boxes represent the 25–75th percentile, center lines represent median values."
- Figures 2C and 4C: No need legends as the x-axis has the labels for each plot.
- Line 31: Report actual p-values instead of saying p<0.05.
- Line 77: Remove 'the' following 'understanding'.
- Line 77: Two periods at the end.
- Line 174: IQR was previously defined.
- Lines 197 and 199: Replace H/E with H&E for consistency throughout the paper.
- Line 303: TPN was defined before.
- Line 334: FXR was defined before.
- Line 356: Correct 'Figure 6' as there is no Figure 6 in the paper.
Author Response
Apr 30, 2020
Editorial Office, Nutrients,
Dear Reviewers,
Thank you once again for considering our manuscript entitled, ‘Impaired gut-systemic signaling drives Total Parenteral Nutrition associated injury’.
Thank you very much for thoroughly critiquing our manuscript. Your recommendations have been extremely helpful. We humbly believe that we have addressed all the critique from reviewers.
Please find below a point by point response to the reviewer comments. Each critique is followed by ‘Response’ in red indicating our action.
We have updated the text in the manuscript per reviewer critique. The entire manuscript has been checked for syntax and grammatical errors. Several new references have been added. Figure legends have been updated. A new table has been added. Additionally, all figures and tables have been revised per journal formatting.
If any section or subsection needs further revision, please let us know and we shall promptly comply.
Sincerely,
Ajay Kumar Jain, MD
******** REVIEWER 2 ********
The study investigates the underlying reasoning of side effects of total parenteral nutrition on the liver and gut by utilizing pig models. The study design seems scientifically accurate and the manuscript was satisfactorily well written. Moreover, the authors report very interesting findings regarding the alterations in gut-derived signaling, which will potentially interest the readers.
I have no major concerns and recommend for acceptance after correcting the following minor issues:
Response: Thank you for recognizing our work. We have addressed the minor issues below.
Critique: Minor:
What are the sizes of the histology images?
Response: Thank you for the critique. While the reviewer notes this as a minor issue, we have added the magnifications. Cholestatic deposits, muscularis mucosa assessment, immune-histochemistry were at 40X magnification. The depicted V/C ratios were at 20X magnification. The Electron microscopy was at 5000X magnification. These have been added to the figure legends.
Critique: Minor
Please use consistent decimals for the p-values reported (preferably 3 decimals for all)
Response: Thank you for the critique. P-values have been reported to 3 decimal places throughout the manuscript as per reviewer critique.
Critique: Minor
What do the asterisks in the boxplot figures mean? I suppose they represent significant differences between the groups. If so, what does the asterisk on the left (near y-axis) mean in Figure 2C?
Response: Thank you for the critique. The reviewer is correct that the asterisks represent significant differences. The asterisks in figure 2C near the y-axis was an error as another asterisk was already present in the TPN group. This has been corrected.
Critique: Minor
No need to explain details of box plots in every figure e.g., "the figure shows box and whisker plots in which boxes represent the 25–75th percentile, center lines represent median values."
Response: Thank you for the critique. We agree that this statement may not need to be repeated. We have followed the journal figure guidelines for legends and most humbly agree for its removal if agreeable by the journal and other reviewers.
Critique: Minor
Figures 2C and 4C: No need legends as the x-axis has the labels for each plot.
Response: Thank you for the critique. We agree. We have followed the journal figure guidelines for legends and most humbly agree for its removal if agreeable by the journal and other reviewers.
Critique: Minor
Line 31: Report actual p-values instead of saying p<0.05.
Response: Thank you for the critique. We have removed the p<0.05 and reported actual p values as per reviewer critique.
Critique: Minor
Line 77: Remove 'the' following 'understanding'.
Response: Thank you for the critique. We have removed ‘the’ following ‘understanding’ per reviewer critique.
Critique: Minor
Line 77: Two periods at the end.
Response: Thank you for the critique. The extra period has been removed per reviewer critique.
Critique: Minor
Line 174: IQR was previously defined.
Response: Thank you for the critique. We have removed ‘interquartile range’ and replaced with IQR per reviewer critique, due to its being defined earlier.
Critique: Minor
Lines 197 and 199: Replace H/E with H&E for consistency throughout the paper.
Response: Thank you for the critique. We have replaced H/E with H&E per reviewer critique.
Critique: Minor
Line 303: TPN was defined before.
Response: Thank you for the critique. We have removed ‘total parenteral nutrition’ and replaced with TPN per reviewer critique, due to its being defined earlier.
Critique: Minor
Line 334: FXR was defined before.
Response: Thank you for the critique. We have removed ‘farnesoid X receptor’ and replaced with FXR per reviewer critique, due to its being defined earlier.
Critique: Minor
Line 356: Correct 'Figure 6' as there is no Figure 6 in the paper.
Response: Thank you for the critique. We noted this error and Figure 6 has been removed per reviewer critique.

Reviewer 3 Report
The authors show with a model in pigs the early effects of TPN versus EN. The research is sound, the analysis is good. Some minor remarks:
- Figure 6 mentioned in the discussion is missing ( Graphic abstract?)
- Since this is performed in piglets is this model also representative for adults? Do developmental issues play a role? Please add a few sentences towards this issue in the discussion.
- mRNA levels for several factors in the gut are lower in the TPN piglets, is this not the effect of the difference in structure , since the effect in the gut is in morphology so much stronger. It would be nice to have immunohistochemistry for protein expression for some of these factors, or otherwise some discussion.
Author Response
Apr 30, 2020
Editorial Office, Nutrients,
Dear Reviewers,
Thank you once again for considering our manuscript entitled, ‘Impaired gut-systemic signaling drives Total Parenteral Nutrition associated injury’.
Thank you very much for thoroughly critiquing our manuscript. Your recommendations have been extremely helpful. We humbly believe that we have addressed all the critique from reviewers.
Please find below a point by point response to the reviewer comments. Each critique is followed by ‘Response’ in red indicating our action.
We have updated the text in the manuscript per reviewer critique. The entire manuscript has been checked for syntax and grammatical errors. Several new references have been added. Figure legends have been updated. A new table has been added. Additionally, all figures and tables have been revised per journal formatting.
If any section or subsection needs further revision, please let us know and we shall promptly comply.
Sincerely,
Ajay Kumar Jain, MD
******** REVIEWER 3 ********
The authors show with a model in pigs the early effects of TPN versus EN. The research is sound, the analysis is good. Some minor remarks:
Response: Thank you for recognizing our work. We have addressed the minor issues below.
Critique: Minor
Figure 6 mentioned in the discussion is missing ( Graphic abstract?)
Response: Thank you for the critique. It does indicate graphical abstract. We noted this error and the work Figure 6 has been removed.
Critique: Minor
Since this is performed in piglets is this model also representative for adults? Do developmental issues play a role? Please add a few sentences towards this issue in the discussion.
Response: Thank you for the critique. We have addressed the relevance of the using the piglet model and its representation for human adult and pediatric studies. We have thus added to the discussion, ‘In this study we have used a novel ambulatory piglet TPN model. We utilized this model as there has been a significant increase in the use of pig as a model to study human physiology largely due to extensive homology of the liver and gastrointestinal tract in both form and function making this highly relevant to human pediatric and adult TPN injury….’
Critique: Minor
mRNA levels for several factors in the gut are lower in the TPN piglets, is this not the effect of the difference in structure, since the effect in the gut is in morphology so much stronger. It would be nice to have immunohistochemistry for protein expression for some of these factors, or otherwise some discussion.
Response: Thank you for the critique. This is an interesting thought. While it is possible that the changes in gut mRNA expression with TPN are a surrogate of the changes in gut structure the same cannot be said with confidence as we are measuring the fold change in expression. Upon tissue grinding for analysis we do normalize the content between groups prior to PCR. We do have IHC noted for downstream signaling that have effects in the liver due to changes in gut receptor and transported activation. One could suppose that in future studies we could plan for gut / enterocyte staining for proliferation (like BrDU) but this is done via an injection when the animal is alive to increase uptake. This would certainly be something we shall plan in future studies. Thanks for this valuable insight. Nevertheless, as per reviewer critique a statement reflecting this has been added to the discussion.

Round 2
Reviewer 1 Report
Thank you for the opportunity to review this work. The author's effort in revising the manuscript must be acknowledged.
However, there are minor comments that I am hoping to seek further clarifications: 1. It would be clearer to the reader if the aim/primary and secondary objectives of the study are clearly written 2. The sample size calculation should be clarified.
Is the sample size calculated based on the change in serum bilirubin or changes in villous to crypt ratio?
And then, what is the statistical test used to analyse this primary outcome? 3. The nutritional content of TPN and EN
- suggest moving to supplementary material 4. The reporting of result should be coherent. For example, if serum bilirubin is going to be reported, it should have been stated under the method section that serum bilirubin will be measured. 5. The author should try to be consistent in the reporting of all the results with values in addition to p-value.
Eg: In page 6 line 204: " We report a significant elevation in serum bilirubin (Figure 1A) in 203 animals on TPN (p=0.034)"
- the level of serum bilirubin should be reported (xx vs xx, p=0.034).
Author Response
May 11, 2020
Editorial Office, Nutrients,
RE: Round 2 of Review
Dear Reviewers,
Thank you once again for considering our manuscript entitled, ‘Impaired gut-systemic signaling drives Total Parenteral Nutrition associated injury’.
Thank you very much for again thoroughly critiquing our manuscript during this second round of review. Your recommendations have been extremely helpful. We humbly believe that we have addressed all the critique.
Please find below a point by point response to the reviewer comments (below). Each critique is followed by ‘Response’ in red indicating our action.
We have updated the text in the manuscript per reviewer critique. The entire manuscript has been again checked for syntax and grammatical errors. Several new references have been added. Additionally, all figures and tables have been revised per journal formatting.
If any section or subsection needs further revision, please let us know and we shall promptly comply.
Sincerely,
Ajay Kumar Jain, MD
Associate Professor, Pediatric Gastroenterology, Hepatology and Nutrition,
SSM Cardinal Glennon Children's Medical Center,
Saint Louis University, 1465 S Grand Blvd,
Tel: 314-577-5647; Email: ajay.jain@health.slu.edu
******Reviewer 1******
Thank you for the opportunity to review this work. The author's effort in revising the manuscript must be acknowledged. However, there are minor comments that I am hoping to seek further clarifications:
Response: Thank you for recognizing our work.
Minor Critique: It would be clearer to the reader if the aim/primary and secondary objectives of the study are clearly written
Response: Thank you for the minor critique. This work describes a novel idea of altered gut-systemic signaling with TPN in contrast to the traditional understanding predicating TPN injury to a toxic or other effect. As per reviewer’s minor critique, we have included an aims statement in the introduction and also included that our key surrogates for TPN injury were serum bilirubin and gut atrophy. We would be happy to make additional changes as requested by the editorial board and the reviewers.
Minor Critique: The sample size calculation should be clarified. Is the sample size calculated based on the change in serum bilirubin or changes in villous to crypt ratio? And then, what is the statistical test used to analyze this primary outcome?
Response: Thank you for the critique. In response to this minor critique we have updated the sample size calculation segment. We had used historical data for a 5-10-fold increase in serum bilirubin and a 2-3 fold decrease in Villous / Crypt ratio for calculating the sample sizes. If only serum bilirubin is taken into consideration, we would have obtained a sample size of 5 per group. We estimated a sample size of 8 animals per group using both these parameters. A Mann Whitney U statistical test was conducted to analyze differences in the primary/key outcomes.
While this is a minor critique as indicated by the reviewer, if necessitated by the editorial, we can provide additional information, however clearly our animal numbers were adequate to find differences in key parameters. We have also included additional references, to this segment indicating our methodology for such calculations.
Minor Critique: The nutritional content of TPN and EN - suggest moving to supplementary material
Response: Thank you for this critique. We had been asked by another reviewer to include the nutritional content in the main manuscript and thus this was added. If the editorial and the other reviewers agree we would be happy to move this table to supplementary material in compliance with this minor critique.
Minor Critique: The reporting of result should be coherent. For example, if serum bilirubin is going to be reported, it should have been stated under the method section that serum bilirubin will be measured.
Response: Thank you for this critique. We have again gone through the manuscript and made sure that the results were reported in a similar manner. As per the reviewer critique, we have made changes to the methods section and noted that a blood sample was obtained just prior to euthanasia and that the samples in each group were analyzed by an automated analyzer at Saint Louis University (SLU) core facility.
Minor Critique: The author should try to be consistent in the reporting of all the results with values in addition to p-value. Eg: In page 6 line 204: " We report a significant elevation in serum bilirubin (Figure 1A) in 203 animals on TPN (p=0.034)" - the level of serum bilirubin should be reported (xx vs xx, p=0.034).
Response: Thank you for this critique. We have reviewed the manuscript. Other reviewers had suggested inclusion of the median and interquartile separately for each group and thus the current format was adopted and continued consistently throughout the manuscript. We have made sure that the same methodology is used throughout the manuscript but would be happy to make additional formatting changes per editorial and reviewer’s decisions.
